# Levels of Aflatoxin M1 in Breast Milk of Lactating Mothers in Monterrey, Mexico: Exposure and Health Risk Assessment of Newborns

**DOI:** 10.3390/toxins14030194

**Published:** 2022-03-05

**Authors:** Rogelio Salas, Nallely Acosta, Aurora de Jesús Garza, Alexandra Tijerina, Roberto Dávila, Zacarías Jiménez-Salas, Laura Otero, Mirna Santos, Antonio-José Trujillo

**Affiliations:** 1Centro de Investigación en Nutrición y Salud Pública (CINSP), Facultad de Salud Pública y Nutrición, Universidad Autónoma de Nuevo León, Monterrey 64460, Mexico; nallely.acostae@uanl.edu.mx (N.A.); alexandra.tijerinas@uanl.mx (A.T.); zacarias.jimenezsl@uanl.edu.mx (Z.J.-S.); laura.oterobtst@uanl.edu.mx (L.O.); mirna.santosl@uanl.mx (M.S.); 2Department of Biochemistry and Molecular Medicine, Facultad de Medicina, Universidad Autónoma de Nuevo León, Monterrey 64460, Mexico; aurora.garzajr@uanl.edu.mx (A.d.J.G.); roberto.davilacnl@uanl.edu.mx (R.D.); 3Centre d’Innovació, Recerca i Transferència en Tecnologia dels Aliments (CIRTTA), TECNIO-UAB, XIA, Departament de Ciència Animal i dels Aliments, Facultat de Veterinària, Universitat Autònoma de Barcelona, 08193 Bellaterra, Spain; toni.trujillo@uab.es

**Keywords:** aflatoxin M1, breast milk, AFM1 daily intake, infant population, carcinogenic risk, Monterrey (Mexico)

## Abstract

The present study aimed to determine the presence of the aflatoxin M1 (AFM1) in breast milk samples from 123 nursing women and the degree of exposure of infants to this toxin, in the metropolitan area of Monterrey, Nuevo Leon state (northeast Mexico). Upon analysis, 100% of the samples were found to be contaminated with the toxin at an average concentration of 17.04 ng/L, with a range of 5.00 to 66.23 ng/L. A total of 13.01% of the breast milk samples exceeded the regulatory limit of 25 ng/L for AFM1 concentration, set by the European Union. The estimated daily intake for AFM1 and the carcinogenic risk index were also determined in the 0- to 6-, 7- to 12-, 13- to 24-, and 25- to 36-month-old age groups. The AFM1 intake through breast milk ranged from 1.09 to 20.17 ng/kg weight/day, and was higher than the tolerable daily intake, indicating a carcinogenic risk for infants in the age groups of 0- to 24-months old. This evidence demonstrates a susceptibility of breast milk to AFM1 contamination that may suggest a carcinogenic risk for the breastfed infants in Monterrey city, Nuevo Leon state, and the need to control the presence of aflatoxins in foods eaten by nursing mothers.

## 1. Introduction

Breast milk is considered as the best food for neonates and infants [1], being a complex biofluid rich in nutrients (e.g., macronutrients, micronutrients, trace elements, etc.) and non-nutritive bioactive components (e.g., cells, immunoglobulins, cytokines, chemokines, hormones, growth factors, etc.) [2,3]. Breast milk contains a variety of bioactive compounds that contribute to proper brain, intestinal, and immune development [4]. The World Health Organization recommends exclusive breastfeeding for the first six months of life, continued until two years of life or longer [5]. 

Breast milk exhibits composition variability among individuals, and it is possible to find the presence of compounds that harm the health of the infant, such as aflatoxins [6]. Aflatoxins (AF) are considered toxic metabolites produced by some molds such as *Aspergillus flavus*, *A. parasiticus* and *A. nomius* [7]. In humans, ingestion of the secondary metabolite AFB1 can lead to the accumulation and excretion of the aflatoxin M1 (AFM1) in breast milk through the ingestion of contaminated food sources, such as corn and derived products, nuts, dairy, sunflower oil, bread, and whole grain cereals [8,9].

AFM1 has been associated with carcinogenic, mutagenic, teratogenic, estrogenic, immunotoxic, nephrotoxic, and neurotoxic effects [8]. In addition, exposure to AFM1 in children has been associated with Reye’s and Kwashiorkor’s syndromes, immunosuppression, dermal irritation, endocrine disruptions, acute hepatitis, and others metabolic disturbances such as non-alcoholic liver disease [10,11,12]. Aflatoxin toxicity is related to intestinal barrier function [13,14]. It has been reported that when aflatoxin is present, there is a disruption of the intestinal barrier, which contributes to oxidative stress and DNA damage [14], and to an inflammatory response [13,14], causing a risk to health.

At present, there is limited scientific evidence on the presence of AFM1 in the breast milk of nursing mothers that have probably been exposed to diets containing AFB1 in Mexico, and the hazards of its presence. The aim of this study was to determine the occurrence and levels of AFM1 in breast milk, the exposure of infants to AFM1, and the potential carcinogenic risks associated with its consumption in the Monterrey population.

## 2. Results and Discussion

This is a first study to demonstrate the presence and concentration of AFM1 in breast milk from mothers in the northeast of Mexico, Nuevo Leon state. In terms of the presence of AFM1 in breast milk, the toxin was detected in 100% of the studied samples (*n* = 123), similar to the results reported from previous studies in other countries [15]. The presence of AFM1 in breast milk is directly related to the consumption habits of lactating mothers due to the pattern of foods they eat, which are different depending on the country. Diets composed of cereals, spices, seeds, oils and nuts, and cow-milk products can be contaminated with AFB1 and AFM1 to a greater or lesser extent depending on several environmental, storage, and climatic conditions that may affect AF production and the contamination of foods with AFB1 [16].

The current literature reports a highly variable occurrence and concentration of AFM1 in breast milk, with certain studies presenting high incidences (100%) such as Gambia and Tanzania [17], Iran [18], the Ecuadorian highlands [19], and Jordan [20], while medium or low occurrences have been reported in Portugal (33%), Ghana (22%), Italy and Cameroon (5%), and Brazil (4%) [21,22,23,24].

Table 1 shows the average, and minimum and maximum AFM1 concentrations found breast milk samples, according to the infant age group. The average concentration of AFM1 in breast milk samples was 17.04 ± 9.13 ng/L and ranged from 5.00 to 66.23 ng/L. Although the average AFM1 concentration in the analyzed breast milk samples did not exceed the tolerance limit of 25 ng/L set by the European Commission for infant formula [25], 13.01% of the individual samples exceeded this limit. On the other hand, it was observed that the breast milk samples collected from nursing mothers at different stages of lactation (age groups) showed similar AFM1 contents in the groups from 0 to 24 months-old, while the AFM1 content of breast milk in the group from 25 to 36 months-old had the lower mean level, although these differences were not significant.

The mean AFM1 levels detected in this study were among of those previously reported in other countries with moderate AFM1 concentrations, such as Turkey (32.83 ng/L), Nigeria (22.33 ng/L), Italy (12.98 ng/L), Brazil (12.01 ng/L) [26], Kuwait (9.7 ng/L) [27], Cyprus (7.84 ng/L) [28], Portugal (7.12 ng/L) [29], Iran (6.80 ng/L) [30], Gambia (5.65 ng/L), Colombia (5.20 ng/L) [31], and Lebanon (4.31 ng/L) [32], but far from the high concentration levels reported by different studies in the United Arab Emirates (210–4060 ng/L) [33], Egypt (7100 ng/L) [34], Sudan (401 ng/L) [35], and Serbia (370 ng/L) [36]. In a study from central Mexico, 89% of breast milk samples (*n* = 112) showed the presence of AFM1 at a concentration of 10.35 ng/L (range of 3.01–34.24 ng/L), and 7% of samples exceeded the recommended limit of 25 ng/L [9].

Table 2 shows the estimated daily intake (EDI; means, and minimum and maximum) of AFM1 through breast milk, and the potential carcinogenic risk index (CRI) according to the equation proposed by Kuiper–Goodman (1994) [37]. Considering the mean AFM1 levels, body weight, and milk intakes of each age group of infants, the EDI values ranged from 1.81 to 5.08 ng/kg weight/day. The youngest group, from 0- to 6-months old, had the highest AFM1 EDI values, showing the susceptibility of this group of infants; EDI values slighly diminished in the subsequent age groups, with the infant group of 25- to 36-months old having the lowest exposure, due to a higher body weight and a lower consumption of breast milk.

In the group aged 0- to 6-months old, infants were exclusively breastfed. However, as infant age increased, there was an introduction of complementary foods such as solid foods, infant formula, liquid milk, or dairy products that may contribute to AFM1 intake. To understand the impact of this, studies carried out in Nuevo Leon, Mexico (northeast region), analyzed several commercial follow-on infant formula and liquid milks (national origin), reporting AFM1 mean concentrations of 180 ng/L (range: 0–450 ng/L) and 520 ng/L (range: 100–1270 ng/L) for these dairy products, respectively [39,40], showing its relevance as a health risk when an infant diet is supplemented with these dairy products.

Limited studies exist estimating the daily intake of AFM1 in infants during the lactation period. The results obtained in this study are close to the EDI values found in Central Mexico (0.92–6.28 ng/kg weight/day) [38] and higher than those found in Lebanon (0.65–0.80 ng/kg weight/day) [33] or Morocco (0.35 ng/kg weight/day) [39], but lower than those recorded in Egypt (52.68 ng/kg weight/day) [41] or Tanzania (11.08 ng/kg weight/day) [42].

The AFM1 intake was higher than the CRI of 2 ng/kg weight/day in infants aged 0- to 24-months old, indicating a high risk of AFM1 exposure for the infant population of northeast Mexico.

These study results reflect the possible mother-to-infant contamination occurring through breast milk, thus increasing the risk by exposing the infants to AFM1 and possibly to other diverse carcinogenic toxins; therefore, it is imperative to establish health regulations to guarantee the quality and safety of food products that may be contaminated by aflatoxins.

## 3. Conclusions

Breast milk samples analyzed from lactating mothers in Monterrey, Nuevo Leon, Mexico, have shown a high occurrence of AFM1 (100% of the samples) with different levels from 5.00 to 66.23 ng/L, in 123 breast milk samples, with 13.01% of samples showing AFM1 levels higher than the upper limits (25 ng/L) established by the EU. Based on the average infant’s body weight and average breast milk intake, the daily AFM1 intake and carcinogenic risk index were estimated in infants aged from 0- to 36-months old, with infants aged from 0- to 24-months old determined to be at risk. These data indicate that lactating mothers in northeast Mexico have a high exposure to AFB1 and/or AFM1 through diet. This evidence strongly supports the need to establish public health programs to give nutrition education to women before and during breastfeeding periods, such that nursing women understand the importance of selecting food choices that lower susceptibility to AFM1 contamination and minimize its passage into breast milk.

## 4. Materials and Methods

### 4.1. Breast Milk Samples

Breast milk samples (*n* = 123) were obtained from breastfeeding women at regular medical visits to local health centers from the metropolitan area of Monterrey city, Nuevo Leon state (northeast region of Mexico), during the summer of 2021. Breast milk samples were collected under hygienic conditions in sterile glass bottles, protected from light, and stored until use at −18 °C. Before analysis, breast milk samples were thawed overnight at 4 °C, and samples were centrifuged at 5000 rpm for 10 min. The upper fatty layer was discarded and 100 μL from each defatted milk sample was used in ELISA assay. All the procedures followed in this study were executed according to the NOM-087-ECOL-SSA1-2002, which establishes the classification of hazardous biological-infectious waste and the specifications of its management [43].

### 4.2. Quantitative AFM1 Determination by ELISA Assay

AFM1 was detected via an ELISA assay, using the RIDASCREEN^®^ kit Aflatoxin M1 (Art. No.: R1121, R-Biopharm AG, Darmstadt, Germany) based on antigen-antibody reactions, as Quevedo-Garza et al. (2018) has described. A 6-point calibration curve was performed in triplicate, with concentrations ranging from 0 to 80 ng/L (0, 5, 10, 20, 40, and 80 ng/L), using certified AFM1 standards. According to the manufacturer’s description, the limit of detection was 5 ng/kg. The resulting data were processed and calculated with the software RIDASOFT^®^Win.NET (v1.96) and reported as nanograms per liter (ng/L).

### 4.3. Estimation of AFM1 Exposure in Infants

To determine the exposure of the infant population to AFM1, the estimated daily intake (EDI) of AFM1 was calculated according to the equation proposed by Quevedo-Garza et al. (2020). Infants were grouped according to their age: 0- to 6-months old, 7- to 12-months old, 13- to 24-months old and 25- to 36-months old. The average breast milk intake and average body weight were obtained from data corresponding to the state of Nuevo León from the National Health and Nutrition Survey (ENSANUT) reported in 2018 [44].

According to the infants’ age groups, milk intake was multiplied by the AFM1 concentration in breast milk, and then divided by infants’ body weight, as stated in the following formula:$$\mathrm{Estimated}\;AFM1\;\mathrm{daily}\;\mathrm{intake}\;(\mathrm{ng}/\mathrm{kg}\;\mathrm{weight}/\mathrm{day})=\frac{\;\left[AFM1\;\left(\frac{\mathrm{ng}}{L}\right)\right]\times \left[Milk\;intake\;\left(L\right)\right]\;}{Body\;weight\;\left(\mathrm{kg}\right)}$$
where *AFM*1 is the average of *AFM*1 concentration in the analyzed samples, expressed in nanograms per liter (ng/L); milk intake is the average daily breast milk intake by the infant population, expressed in liters (L); and body weight is the average body weight of the infants from each age group, expressed in kilograms (kg).

### 4.4. Carcinogenic Risk Index

A carcinogenic risk index (CRI) was established for the infants consuming AFM1 through breast milk by comparing the levels of estimated AFM1 daily intake (EDI) vs. the tolerable daily intake (TDI) of 2 ng/kg weight/day, as previously reported by Kuiper–Goodman (1994) and Quevedo-Garza et al. (2020).

### 4.5. Statistical Analysis

All samples were analyzed in duplicate and the AFM1 concentrations in breast milk were expressed as mean ± standard deviation (SD), with minimum and maximum values. In order to evaluate the AFM1 concentrations within the different age-groups, one-way ANOVA was performed using the SPSS^®^ v17.0 package to a 95% level of significance, and Tukey adjustment was performed (*p* < 0.05).

## Figures and Tables

**Table 1 toxins-14-00194-t001:** Aflatoxin M1 (AFM1) presence in breast milk.

Age Group (Months)	*n* (%) *	AFM1 (ng/L)	
		Mean ± SD	Min–Max
0 to 6	86 (9.30)	16.68 ± 9.66	5.00–66.23
7 to 12	18 (22.22)	17.92 ± 8.45	5.00–38.26
13 to 24	16 (25.00)	18.96 ± 7.49	5.00–29.21
25 to 36	3 (00.00)	11.96 ± 3.46	8.16–14.93
Total	123 (13.01)	17.04 ± 9.13	5.00–66.23

* *n* is the number of samples of breast milk collected for each age group. The value in parentheses indicates the samples’ percentage above the limit set by the European Union (25 ng/L) with respect to the total.

**Table 2 toxins-14-00194-t002:** Estimated daily intake (EDI) of aflatoxin M1 and the carcinogenic risk index (CRI) according to infant age group.

Age Group (Months)	Body Weight (kg)	Milk Intake (L/day)	EDI (ng/kg Weight/Day)		CRI * (2 ng/kg Weight/Day)
			Mean	Min–Max	
0 to 6	6.50	1.98	5.08 ± 2.94	1.52–20.18	Risk
7 to 12	8.99	2.35	4.68 ± 2.21	1.31–9.98	Risk
13 to 24	10.93	2.37	4.10 ± 1.62	1.08–6.33	Risk
25 to 36	13.24	2.02	1.81 ± 0.52	1.25–2.28	Potential risk

* CRI: carcinogenic risk index, as reported by Kuiper–Goodman (1994) and Quevedo-Garza et al. (2020) [37,38].

## Data Availability

Not applicable.

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
