# Peer review of "Levels of Aflatoxin M1 in Breast Milk of Lactating Mothers in Monterrey, Mexico: Exposure and Health Risk Assessment of Newborns"

_toxins, 2022, doi:10.3390/toxins14030194_

Round 1
Reviewer 1 Report
MANUSCRIPT DETAILS
Ms. Ref. No.: toxins-1610362
Title:
Levels of Aflatoxin M1 in Breast Milk of Lactating Mothers in Monterrey, Mexico: Exposure and Health Risk Assessment of Newborns.
Article Type: Research Article
Journal: Toxins
GENERAL COMMENTS
This manuscript aims to comprehensively provide general information of determine the occurrence and levels of AFM1 in breast milk, the exposure of infants to AFM1 and its potential carcinogenic risk associated with its consumption in the Monterrey population.
The interest in this manuscript is significant enough to merit publication.
My recommendation on submitted manuscript to the Toxins journal is to be accepted after major revisions.
The comments and questions provided below may help the authors to put the manuscript into better appropriate form for publication.
SPECIFIC COMMENTS
Abstract
- Kindly specify clear and concise aim for your work, and the aim of Levels of Aflatoxin M1 in Breast Milk of Lactating Mothers in both “Abstract” and at the end of the “Introduction” sections.
Result and discussion
- Please I would like correlation between Estimation of AFM1 exposure in infants and Carcinogenic risk index
References
- Revise references to be formatted in accordance with Journal of Toxins Author Guidelines.
Author Response
Response to Reviewer 1 Comments
Point 1: GENERAL COMMENTS
This manuscript aims to comprehensively provide general information of determine the occurrence and levels of AFM1 in breast milk, the exposure of infants to AFM1 and its potential carcinogenic risk associated with its consumption in the Monterrey population.
The interest in this manuscript is significant enough to merit publication.
My recommendation on submitted manuscript to the Toxins journal is to be accepted after major revisions.
The comments and questions provided below may help the authors to put the manuscript into better appropriate form for publication.
Response 1: Before responding to your comments, we would like to thank you for your very useful and detailed referee report which has undoubtedly contributed to the improvement of the article. We are very grateful for the time and effort that you have devoted to our paper and we expect that the thorough revision made to the manuscript addresses your concerns.
SPECIFIC COMMENTS
Point 2: Abstract
Kindly specify clear and concise aim for your work, and the aim of Levels of Aflatoxin M1 in Breast Milk of Lactating Mothers in both “Abstract” and at the end of the “Introduction” sections.
Response 2: Thank you for your observation. We have specified clearly the aim of the study in the Abstract section, and we have explaining better the meaning of AFM1 levels of breast milk of lactating mothers in both Abstract and at the end on the Introduction sections.
Result and discussion
Point 3: Please I would like correlation between Estimation of AFM1 exposure in infants and Carcinogenic risk index
Response 3: In this work we have not epidemiological data on aflatoxicoses in infants, so we can not possible make a correlation between epidemiological data and AFM1 consumption. In our study, based on the concentration of AFM1 found in breast milks and the weight of the infants, we have estimated the AFM1 daily intake and compared it with the tolerable daily intake (TDI) of 2 ng/kg weight/day, as previously reported by Kuiper-Goodman (1994) in order to know the carcinogenic risk index.
References
Point 4: Revise references to be formatted in accordance with Journal of Toxins Author Guidelines.
Response 4: Thank you for your observation. We have revised the references format.

Reviewer 2 Report
Levels of Aflatoxin M1 in Breast Milk of Lactating Mothers in Monterrey, Mexico: Exposure and Health Risk Assessment of Newborns
Manuscript submitted to TOXINS
The subject can be of interest for researchers involved in the study of the sources of Aflatoxin M1. Even some papers dealing with the levels of Aflatoxin in breast milk do exist, the content of this report is very engaging and provides some new information, however it has scope for its improvement before its final acceptance. The following are some edits/comments for the authors' consideration to help improve the clarity of the paper.
Introduction
Introduction should be extended. the authors expounded that the sources, endangerment of AFM1. However, the pathogenic mechanism of aflatoxin should be supplemented. Significant literature on mechanism of aflatoxin is not included. Authors are advised to introduce the following studies in the paper.
Gao, Y.; Bao, X.; Meng, L.; Liu, H.; Wang, J.; Zheng, N. Aflatoxin B1 and Aflatoxin M1 Induce Compromised Intestinal Integrity through Clathrin-Mediated Endocytosis. Toxins 2021, 13, 184. https://doi.org/10.3390/toxins1303018
Yang H.; Wang, Y.; Yu, C.; Jiao, Y.; Zhang, R.; Jin, S.; Feng, X. Dietary Resveratrol Alleviates AFB1-Induced Ileum Damage in Ducks via the Nrf2 and NF-κB/NLRP3 Signaling Pathways and CYP1A1/2 Expressions. Agriculture 2022, 12, 54. https://doi.org/10.3390/agriculture12010054
Results and Discussion
Line 85: the word“medium”should be “mean”
Line 89:“The youngest group from 0- to 6-month-old had the highest AFM1 intake”, “intake” should be “EDI”
Materials and Methods
Line 135: Please provide more information about ELISA kits (including article number, brand and origin)
Line 144: Provide the full name of “LOD” when the abbreviation appeared for the first time.
Line 157: The abbreviation of “AFM1”should be in the formula should be ‘AFM1”, and the unit of AFM1 “ng/L” should be added in the formula.
Line 158: “Where AFM1 is the average AMF1 concentration in the analyzed samples”, “AMF1” should be “AFM1”.

Author Response
Response to Reviewer 2 Comments
Point 1: The subject can be of interest for researchers involved in the study of the sources of Aflatoxin M1. Even some papers dealing with the levels of Aflatoxin in breast milk do exist, the content of this report is very engaging and provides some new information, however it has scope for its improvement before its final acceptance. The following are some edits/comments for the authors' consideration to help improve the clarity of the paper.
Response 1: Before responding to your comments, we would like to thank you for your very useful and detailed referee report which has undoubtedly contributed to the improvement of the article. We are very grateful for the time and effort that you have devoted to our paper and we expect that the thorough revision made to the manuscript addresses your concerns.
Point 2: Introduction
Introduction should be extended. the authors expounded that the sources, endangerment of AFM1. However, the pathogenic mechanism of aflatoxin should be supplemented. Significant literature on mechanism of aflatoxin is not included. Authors are advised to introduce the following studies in the paper.
Gao, Y.; Bao, X.; Meng, L.; Liu, H.; Wang, J.; Zheng, N. Aflatoxin B1 and Aflatoxin M1 Induce Compromised Intestinal Integrity through Clathrin-Mediated Endocytosis. Toxins 2021, 13, 184. https://doi.org/10.3390/toxins1303018
Yang H.; Wang, Y.; Yu, C.; Jiao, Y.; Zhang, R.; Jin, S.; Feng, X. Dietary Resveratrol Alleviates AFB1-Induced Ileum Damage in Ducks via the Nrf2 and NF-κB/NLRP3 Signaling Pathways and CYP1A1/2 Expressions. Agriculture 2022, 12, 54. https://doi.org/10.3390/agriculture12010054
Response 2: Thank you for your observation. The Introduction section has been extended including the pathogenic mechanism of aflatoxin and the two references suggested by you.
Point 3. Results and Discussion
Line 85: the word“medium”should be “mean”
Line 89:“The youngest group from 0- to 6-month-old had the highest AFM1 intake”, “intake” should be “EDI”
Response 3: Thank you for your observation. Changes suggested by you have been done.
Point 4: Materials and Methods
Line 135: Please provide more information about ELISA kits (including article number, brand and origin)
Line 144: Provide the full name of “LOD” when the abbreviation appeared for the first time.
Line 157: The abbreviation of “AFM1” should be in the formula should be ‘AFM1”, and the unit of AFM1 “ng/L” should be added in the formula.
Line 158: “Where AFM1 is the average AMF1 concentration in the analyzed samples”, “AMF1” should be “AFM1”.
Response 4: Thank you for your observations. The information about the ELISA kit used in AFM1 determination is in the “5.2. Quantitative AFM1 determination by ELISA assay” section. We have (1) defined LOD, (2) made the change to AFM1 and included its units in the formula, and (3) change AMF1 by AFM1.

Reviewer 3 Report
The manuscript reports the AFM1 occurrence in breast milk from lactating mothers in Monterrey, Nuevo Leon, in Mexico. In some samples, AFM1 levels exceed the tolerance limits of 25 ng/L set by the European Commission for infant formula. Although a large body of literature reports showing AFM1 presence in breast milk, Authors estimated the daily intake of this mycotoxin in infants during the lactation period. The undertaken research topic fits with the scope of the Toxins journal. The paper is well-written and the methods were described correctly. I recommend the manuscript for publication after minor revision.
Please find my additional comments below:
Table 1. Please discuss if there is any difference between AFM1 levels determined in breast milk within studied age groups.
Please keep the same number of decimal places within the text and in tables (lines 74-76).
Author Response
Response to Reviewer 3 Comments
Point 1: The manuscript reports the AFM1 occurrence in breast milk from lactating mothers in Monterrey, Nuevo Leon, in Mexico. In some samples, AFM1 levels exceed the tolerance limits of 25 ng/L set by the European Commission for infant formula. Although a large body of literature reports showing AFM1 presence in breast milk, Authors estimated the daily intake of this mycotoxin in infants during the lactation period. The undertaken research topic fits with the scope of the Toxins journal. The paper is well-written and the methods were described correctly. I recommend the manuscript for publication after minor revision.
Response 1: Before responding to your comments, we would like to thank you for your very useful and detailed referee report which has undoubtedly contributed to the improvement of the article. We are very grateful for the time and effort that you have devoted to our paper and we expect that the thorough revision made to the manuscript addresses your concerns.
Please find my additional comments below:
Point 2: Table 1. Please discuss if there is any difference between AFM1 levels determined in breast milk within studied age groups.
Response 2: Thank you for your observation. A discussion on the AFM1 levels determined in breast milk within studied age groups has been including in the text.
Point 3: Please keep the same number of decimal places within the text and in tables (lines 74-76).
Response 3: Thank you for your observation. The number of decimal places have been homogenized in the text and in tables.

Round 2
Reviewer 1 Report
Accept in present form